# Evaluation of the Precipitation Process of a Clad Pipe by the Thermoelectric Potential Technique

**Ricardo Carabes [1], Héctor Carreón [1,\*], Maria L. Carreon [2], Melchor Salazar [3] and Pedro Hernández [4]**

[1] Instituto de Investigaciones Metalúrgicas (UMSNH), Ciudad Universitaria, Morelia 58000-888, Mexico; ricardogc@iim.unam.mx

[2] Chemical and Biological Engineering Department, South Dakota School of Mines & Technology, Rapid City, SD 57701, USA; Maria.CarreonGarciduenas@sdsmt.edu

[3] Clúster Politécnico Veracruz-IPN, Av. Instituto Politécnico Nacional S/N, Veracruz 93400, Mexico; msalazarm@ipn.mx

[4] Instituto Mexicano del Petróleo, Eje Central Lázaro Cárdenas Norte 152, San Bartolo Atepehuacan GAM CDMX 07730, Mexico; phernand@imp.mx

\* Correspondence: hcarreon@umich.mx; Tel.: +1-52-443-3167414

**Abstract:** The article reports a study carried out on metallic samples extracted from a metallurgically bonded clad pipe (API 5L X65 steel-Inconel 825 alloy) subjected to a solution heat treatment at 1200 °C and a subsequent aging treatment at 650 °C for different times in order to promote microstructural changes in thermo-metallurgical bonded materials. The non-destructive thermoelectric potential (TEP) technique was used to monitor microstructure changes due to the artificial aging process. In addition, micro-hardness tests were carried out on the metallic materials and micrographs were obtained by means of an optical and scanning electron microscope (SEM). The TEP value changed with solution treatment temperature and reached a maximum value for solution treatment at 1200 °C. The changes in TEP during solution treatment were caused by changes in the solubility of the alloying elements. In the artificial aging process, the TEP value decreased with increasing aging time due to the precipitation process, but exhibited distinct characteristics for the different zones at the clad pipe samples.

**Keywords:** clad pipe; hardness; TEP; microstructure; aging

## 1. Introduction

In the oil industry, the pipe system is a fundamental part of the transporting infrastructure, which directly affects the production costs [1]. The depletion of reserves in near-shore waters and the increase in the demand for hydrocarbons have resulted in a global grow of the exploitation at deep waters [2]. Over time, components such as $H_2S$ and $CO_2$ deteriorate the steel pipes because of a natural aging process. This natural process is accelerated due to prolonged exposure at operating temperatures between 25 and 70 °C in addition to variations in the working pressure [1,3]. These conditions induce changes in the material microstructure, mechanical properties (elastic limit, hardness, ductility, and toughness), fracture, and failure after several years of service. On the other hand, clad pipes have been used as a safe method to transport contaminated oil and gas from the well to the processing facilities [3]. The clad pipes consist of a carbon steel base material as the backing material and provide the desired mechanical properties (strength and elongation), and an internal or external corrosion-resistant alloy (CRA) layer, which is in contact with the medium to be transported to provide corrosion and cracking protection. The most common CRA layer thickness is 3 mm, which is sufficient to provide adequate

corrosion protection in harsh service conditions. From a practical point of view, it offers a wide margin of safety for the field welding process [3–6]. One of the main challenges that clad pipes have faced is the detection by nondestructive evaluation (NDE) of possible cracks, or cracks produced either by their use or manufacture, due to the dissimilar nature of both materials. In this research work, the relationship between thermoelectric potential (TEP) and microstructure was obtained using the contact TEP NDE technique (hot tip) and SEM at the clad pipe (API 5L X65/Inconel 825, Japan Steel Works LTD, Muroran, Japan) artificially aged.

The non-destructive thermoelectric technique is based on the Seebeck effect [7]. Figure 1 shows a diagram of the characterization of metallic materials by thermoelectric potential (TEP) means. The thermoelectric voltage measurement is given by $V = \int_{T_c}^{T_h} S_{SR}(T)dT$ where $T_h$ is the preset temperature of the reference electrode, $T_c$ is the room temperature, $S_{SR}$ is the resulting thermoelectric potential given by the absolute thermoelectric powers of the sample and the reference electrode, and $T$ is the temperature. The measurement is performed quickly to ensure that the hot reference electrode does not reduce its temperature and that the rest of the specimen does not heat significantly. The TEP is affected by network defects such as solute atoms, dislocations, and precipitates, which alter the electronic or elastic properties of the materials and, therefore, induce a variation of the TEP measurements [8–10].

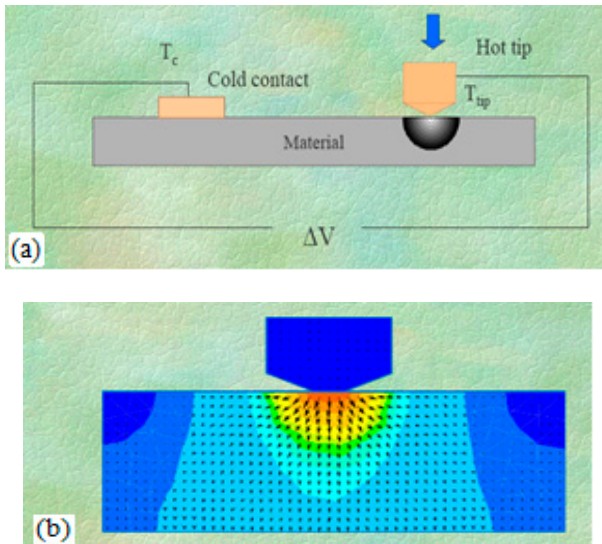

**Figure 1.** Schematic diagram of the (**a**) hot tip thermoelectric method and the (**b**) thermal gradient inside the sample.

## 2. Experimental Method

### 2.1. Material

The base material was API 5L X65 Steel, which is one of the most used materials in the manufacture of clad pipes in the oil industry. The characteristics of this type of micro-alloyed steel are related to the nature of the grain size and the precipitation of carbides and microalloying nitrides. The nature of precipitation, as well as its solubility temperatures, are important factors for the control of the responses in the performance of HSLA (high-strength low-alloy steel) steel. Its chemical composition is illustrated in Table 1. The cladding material was an Inconel 825 alloy, which is a nickel–iron–chromium alloy, with additions of other elements in smaller quantities. The chemical composition of this nickel-based alloy is designed to provide a greater resistance response in highly corrosive environments. Its chemical composition is given in Table 2. Samples were extracted from wall sections of a 24-inch clad pipe manufactured by hot metallurgical bonding process, where a pair of CRA sheets are placed in the middle of base metal sheets, welded together around the edges, and then hot rolled to the required thickness.

**Table 1.** Chemical composition of API 5L X65 micro-alloyed steel (wt%).

| C | Mn | Si | P | S | Al | Nb | Cu | Cr | Ni | V | Ti |
|---|----|----|---|---|----|----|----|----|----|---|----|
| 0.04 | 1.48 | 0.25 | 0.12 | 0.002 | 0.041 | 0.047 | 0.1 | 0.02 | 0.08 | 0.07 | 0.017 |

**Table 2.** Chemical composition of Inconel 825 alloy (wt%).

| Al | Fe | S | C | Mn | Si | Cr | Mo | Ti | Cu | Ni |
|----|----|---|---|----|----|----|----|----|----|----|
| 0.11 | 31.1 | 0.001 | 0.01 | 0.44 | 0.07 | 21.9 | 2.84 | 1.05 | 1.87 | 40.6 |

Clad pipe testing samples were prepared according to Figure 2 and prior heat treatments.

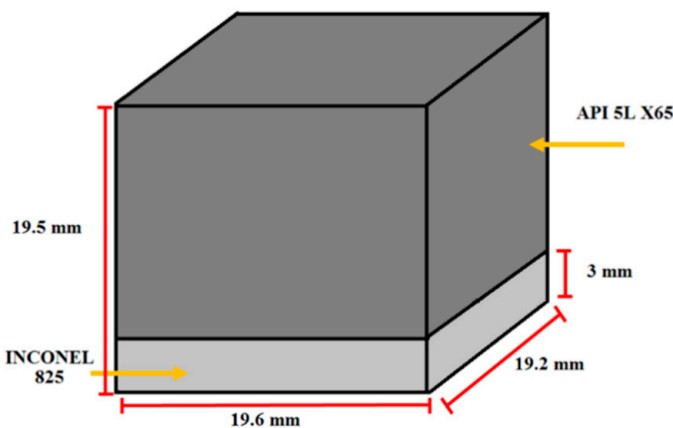

**Figure 2.** Dimensions of the clad pipe samples.

## 2.2. Artificial Aging

A solution heat treatment was carried out at a temperature of 1200 °C for 1 h followed by water cooling/quenching (WQ). Then, an aging heat treatment was performed isothermally at 650 °C at different aging times of 5, 10, 15, 22, 30, 50, 120, 150, and 312 h (Figure 3).

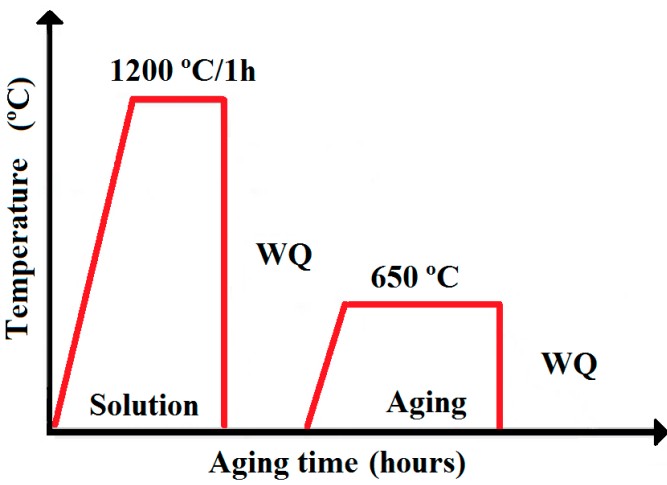

**Figure 3.** Schematic thermal cycles. WQ = water quenching.

## 2.3. Metallographic Characterization

The heat-treated samples were prepared with conventional metallographic techniques for microstructure characterization. API 5L X65 microstructure was revealed with 5% Nital chemical attack using the immersion technique with oscillating movement in the exposed polishing surface

solution. In the case of the Inconel 825 alloy, microstructures were revealed with an electro-chemical attack technique using 8% sulfuric acid diluted in distilled water and 3 volts applied for 10 s. Close examination of revealed microstructures was performed by using optical and scanning electron microscopy (SEM) with energy dispersive spectroscopy (EDS).

*2.4. Vickers Hardness*

The effect of the aging heat treatment on the micro-alloyed X65 steel, Inconel 825 alloy, and the interface were evaluated with hardness measurements using a Mitutoyo Japan HM-200 Vickers hardness tester. The indenter (made of diamond) had a square-base pyramidal geometry with an included angle of 136°. The Vickers hardness number (HV) is the ratio of applied load to the surface area of the indent. According to the Vickers prescription, HV = 1.8544 P/$d^2$ and the result is reported as the average value in units of kg/mm$^2$. Ten indentations were made in each zone per sample with a pyramidal diamond indenter using a load of 100 g for 10 s for the X65 steel [11], 50 g for the Inconel 825 alloy, and 25 g for the interface [12,13], as shown in Figure 4.

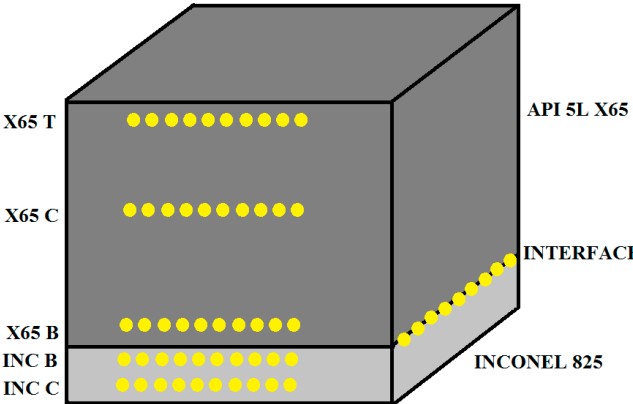

**Figure 4.** Hardness indentation zones at the clad pipe samples: (T) top, (C) center, (B) base, and the interface.

Once the aging heat treatments were finished, the TEP measurements were made using the hot tip TEP method. The thermoelectric instrument included two copper tips at different temperatures. The hot tip was approximately at 53 °C and the cold tip was at room temperature (25 °C), obtaining a temperature differential of 28 °C. The procedure consisted of waiting for the reference temperature of the equipment to stabilize, followed by testing the bonded clad pipe samples. After having the TEP measurements made, an arithmetic average of thirty measurements was obtained, which represented the absolute value of the thermoelectric potential.

## 3. Results and Discussion

*3.1. Metallurgical Characterization*

Figure 5 shows the optical micrographs of the sample with the solution heat treatment (SHT) in the interface. The interface was about 32 μm avg. Also, the presence of Ni, Fe, Mn, Cr, and C were detected by EDS microanalysis.

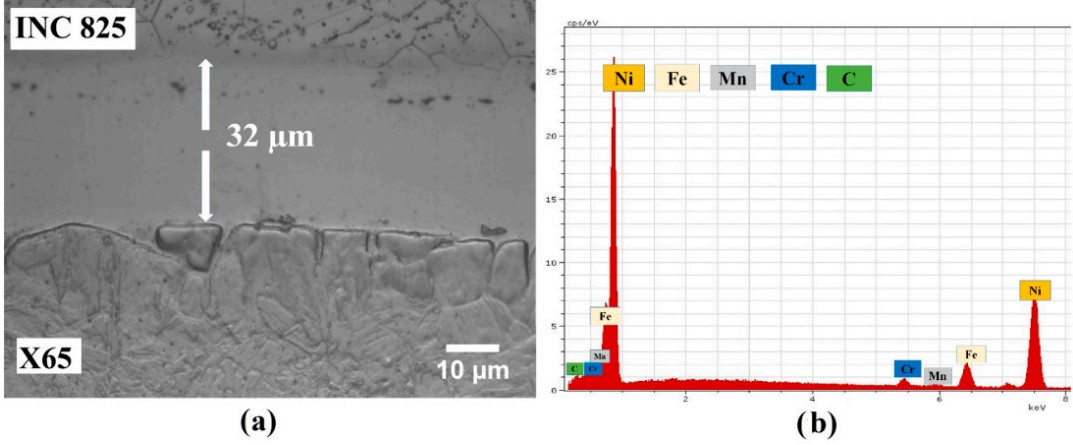

**Figure 5.** Microstructures of the (**a**) solution heat treatment (SHT) sample and (**b**) its EDS microanalysis.

The SHT sample (Figure 6a) presented a grayish tinted band along the length of the API-X65/825 interface, approximately 0.5 μm in thickness. This can be attributed to carbon diffusion influencing the local chemical composition and thus influencing the etching process. The interface of the samples over-aged at 650 °C for three different aging times was dark and pronounced compared to the SHT sample, as seen by comparing Figure 6a–d. The carbon enriched-region spanned 1–2 μm into the clad. This indicates that the aging process works to provide much more diffusion of carbon into the interface.

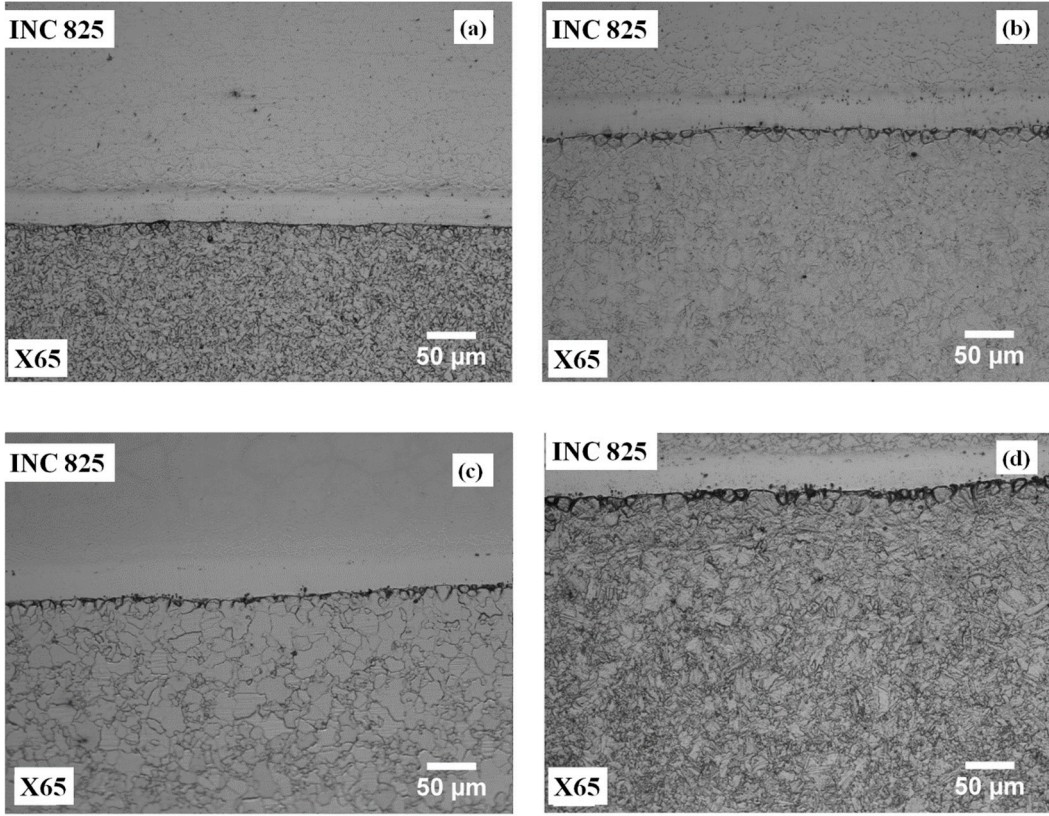

**Figure 6.** Optical micrographs of the clad-base metal interface (**a**) with solution heat treatment (SHT) and over-aged at 650 °C for three different aging times of (**b**) 50 h, (**c**) 150 h, and (**d**) 312 h, respectively.

SEM analysis on the API 5L X65 matrix of clad samples revealed the presence of thin dispersed precipitation of FeC particles detected by EDS microanalysis, showing important changes in the amount, size, and chemical composition as a function of aging time [14]. These nanoprecipitates

showed preferential nucleation and interactions within the ferritic grains, shown in Figure 7. During the early periods of the aging process, a precipitation of fine $Fe_3C$ and $Fe_2C$ precipitates of 50–200 nm in size was presented [4] until peak-aging at 150 h (see Figure 7c), followed by a coarsening process of carbides, which was the dominant microstructural process as a consequence of over-aging at 312 h (see Figure 7d).

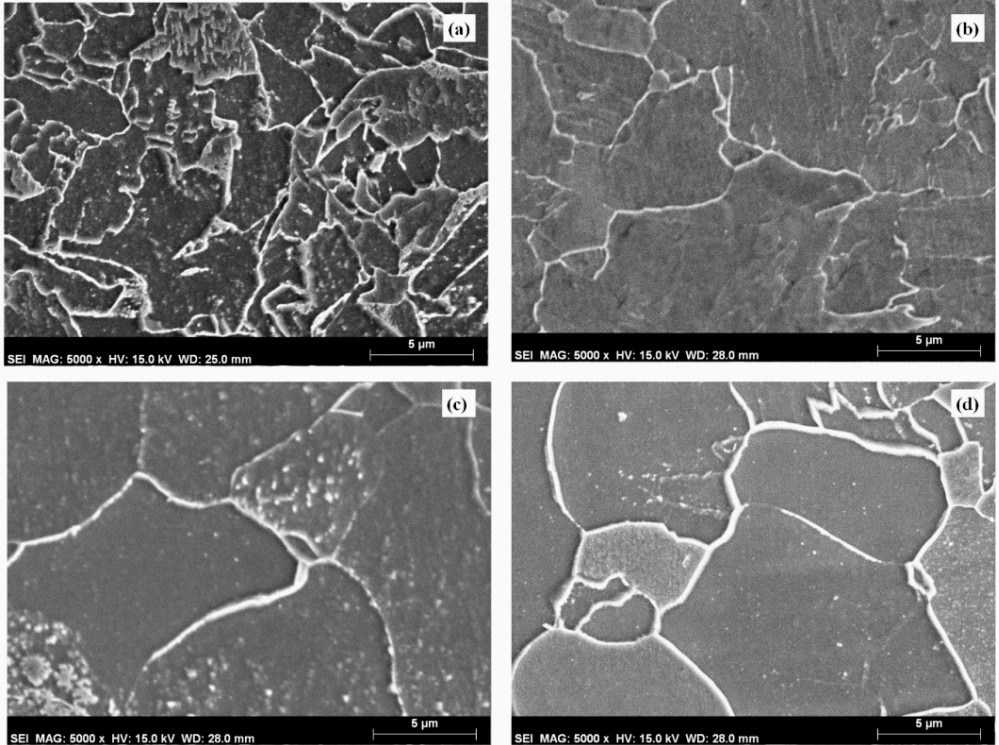

**Figure 7.** SEM micrographs of the micro-alloyed API-X65 steel (**a**) as received and over-aged at 650 °C for three different aging times of (**b**) 50 h, (**c**) 150 h, and (**d**) 312 h, respectively, showing fine $Fe_3C$ and $Fe_2C$ precipitates homogeneously distributed in ferritic grains.

Figure 8 shows the micrographs obtained with the SEM on Inconel 825 alloy matrix, where the presence of precipitates can be observed. The austenite ($\gamma$) matrix represented the highest proportion in this super-alloy, which was composed of a high percentage of Ni and other elements in smaller quantities such as Cr, Mo, and W. The amount of precipitates became more evident as the aging time increased (see Figure 8). $M_{23}C_6$ carbides were the most abundant in the Inconel 825 alloy induced by the aging process. These carbides precipitated preferentially in the grain boundaries and secondly within the matrix. These carbides were mainly composed of Cr and Mo/Ti. Other carbides can be observed in Figure 8, corresponding to the $MC_6$ type. This $MC_6$ carbide type was found in smaller proportions and its precipitation took place in the 825 alloy matrix. The most stable carbide presented was the MC type (see Figure 8d). MC carbide was formed mostly of Ti and precipitate in the matrix and in the grain boundary. This type of carbide is known to increase hardness [15,16].

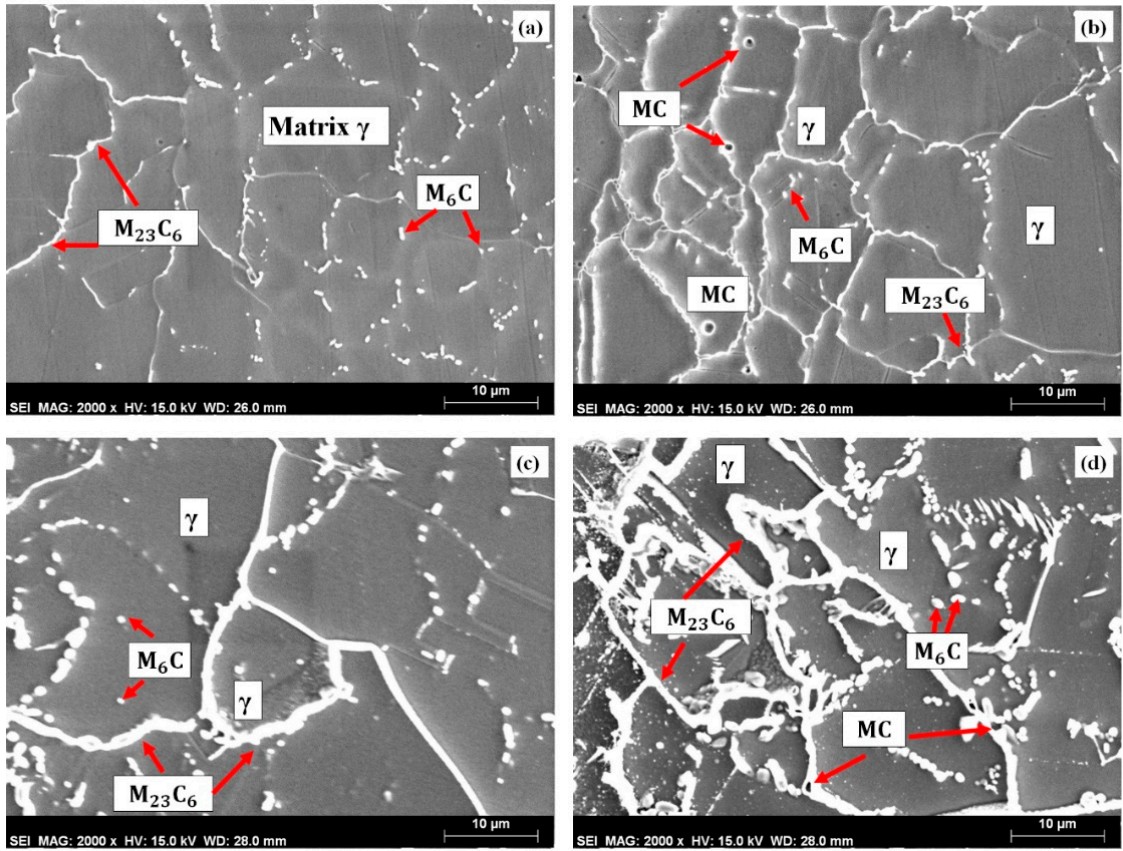

**Figure 8.** SEM micrographs of the Inconel 825 alloy (**a**) as received and over-aged at 650 °C for three different aging times of (**b**) 50 h, (**c**) 150 h, and (**d**) 312 h, respectively.

## 3.2. Hardness Measurements

The results obtained from the Vickers hardness measurements were represented with the aid of the OriginPro 8 program for the different zones and aging times at the clad pipe samples. Figure 9 illustrates the behavior of the hardness data, where it can be seen that important changes occur at distinct zones and aging times. It is clearly observed that the Vickers hardness values for the Inconel 825 alloy show a tendency to increase after 100 h of the aging heat treatment mainly in the base zone. The increase in the Vickers hardness of the Inconel 825 is due to the hardening mechanism caused by dispersion of $MC_6$ carbides and by the formation of $M_{23}C_6$ and MC precipitates. The Vickers hardness data for the micro-alloyed X-65 steel present a trend to increase in the early aging times (precipitation of $Fe_3C$ and $Fe_2C$ carbides) and then a decrease is observed with long times of aging (thickening of carbides). In the interface, the aged samples showed a higher average of clad hardness after 150 h compared to the SHT sample. The high hardness in the bonded clad could indicate the presence of martensite. A high content of alloying elements, especially carbon, which increases hardenability, but also Ni to some extent, may facilitate martensite formation upon cooling from elevated temperatures, as austenite is stabilized over a wider temperature and compositional range [17].

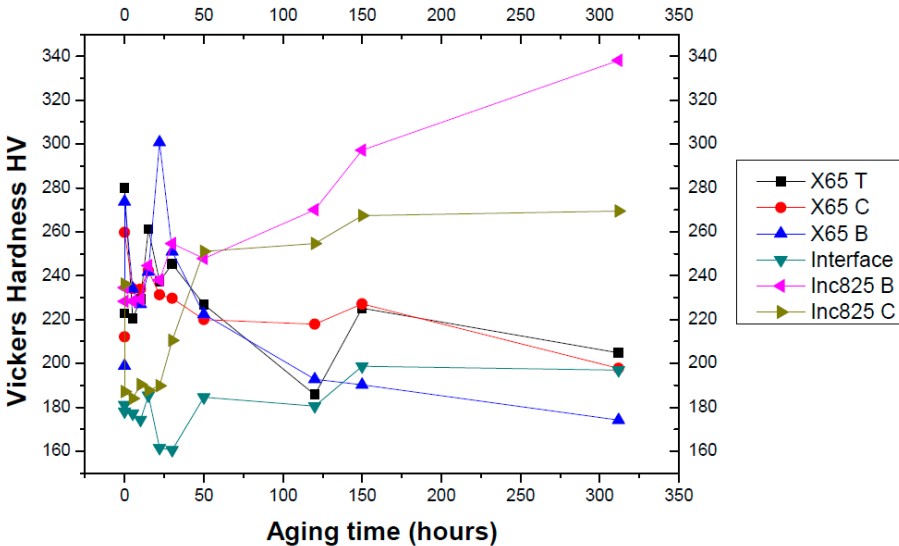

**Figure 9.** Hardness data on the material base of clad pipe samples at distinct zones and aging times.

*3.3. TEP Measurements*

The TEP measurements were plotted with respect to the distinct zones and aging times as shown in Figure 10. It can be seen that the absolute TEP data of the micro-alloyed X65 steel in the different zones is greater than those of the Inconel 825 alloy, which have even negative TEP values in the highest aging times, especially in the central zone. At this point, it can be concluded that, in the first stage of the aging process, the main factors that influenced the decrease of the TEP values were the depletion or enrichment of the matrix from the precipitating elements and the noticeable microstructural changes in the different zones. In the micro-alloyed steel at the bottom (X65B), an equiaxed grain microstructure was presented while, in the center and top zones (X65C and X65T), a mixture of equiaxed and elongated grain was developed. In the Inconel 825 alloy, at the bottom zone (INC B), an equiaxed and small grain was presented while, in the center zone (INC C), a mixture of bigger equiaxed and elongated grain was developed. In the second stage of the aging process, the TEP apparently decreased due to the inhomogeneity of the microstructure (grain size/shape) that induced texture in the medium, which had a negative effect on the TEP and counteracted the precipitation effect due to the aging process induced by the heat treatments, especially in the Inconel 825 alloy center zone. On the other hand, the increased carbon content in the interface zone was due to carbon diffusion decreasing the TEP values [18]. This implies that carbon diffusion induces the formation of a hard, crack susceptible microstructure in the bonded clad.

In order to compare the results of the hardness and TEP measurements, a correlation was carried out between both testing techniques (see Figures 11 and 12) for the different zones and aging times. The comparison in hardness and the absolute TEP at the micro-alloyed X65 steel and interface shows a direct relation of the behavior between them (see Figure 11), where it can be observed as the aging time increases for each zone, both experimental parameters tend to decrease. The hardness shows a peak value at a certain aging time, but the TEP value monotonically decreases as the aging time increases. The TEP values, although they decrease with time, show differences in their rates of change with increasing aging times specially at the interface. In the comparison of the hardness and the absolute TEP of the Inconel 825 alloy (see Figure 12), there is an inverse relationship, which indicates that the carbides and/or precipitates induce hardening and negatively affect the TEP in the material. An increase in the hardness was found as a function of the aging time, which was linked to the precipitation of carbides such as $MC_6$ and $M_{23}C_6$ up to peak-aging at 50 h, and to a coarsening process of these carbides and the formation MC particles due to over-aging after 50 h. The formation of chromium carbides such as $M_{23}C_6$ in the grain boundaries caused areas of chromium impoverishment, which induced defects and inhomogeneities.

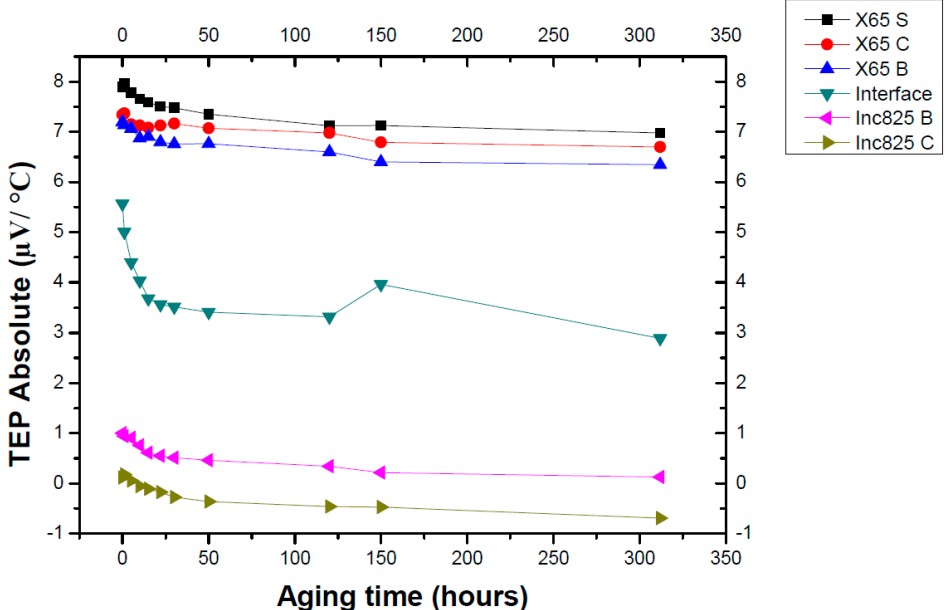

**Figure 10.** Thermoelectric potential (TEP) data of the metal joint in a clad pipe sample artificially aged at different zones.

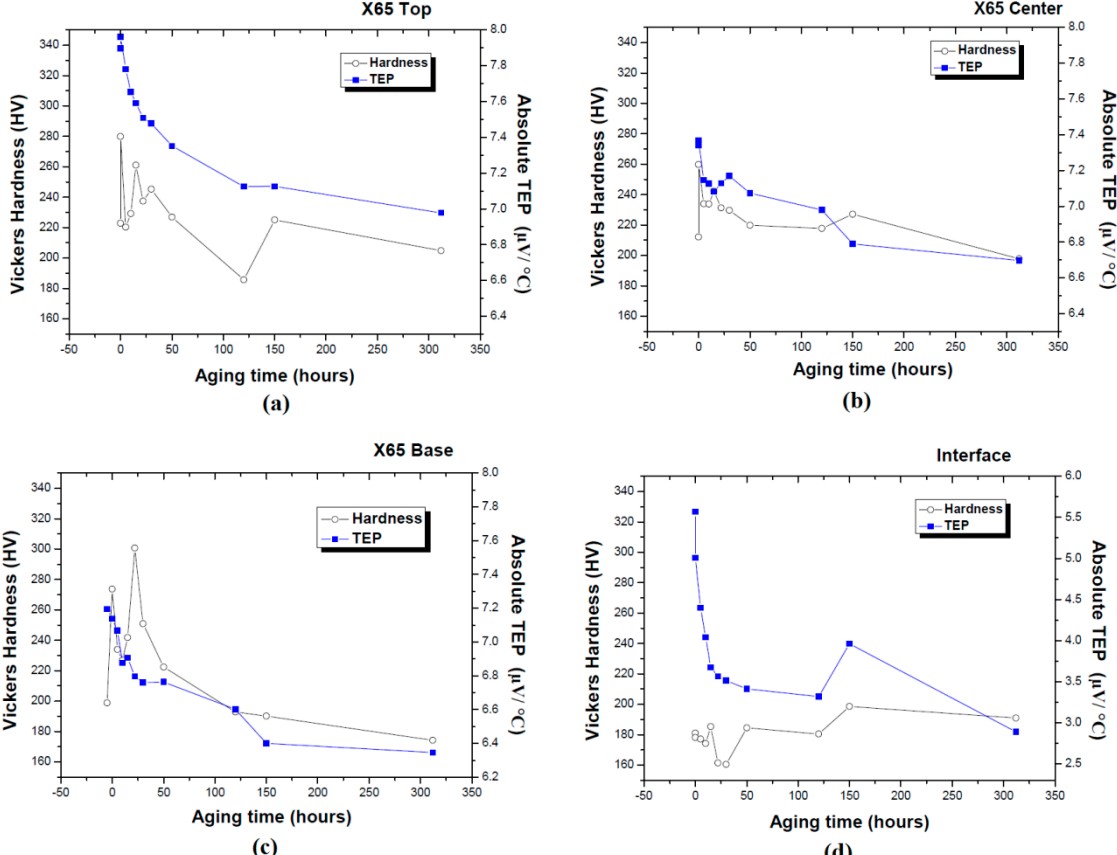

**Figure 11.** Graphs for comparison of Vickers hardness vs. aging time in the micro-alloyed X65 steel for (**a**) the top zone, (**b**) the center zone, (**c**) the base zone, and (**d**) the interface zone.

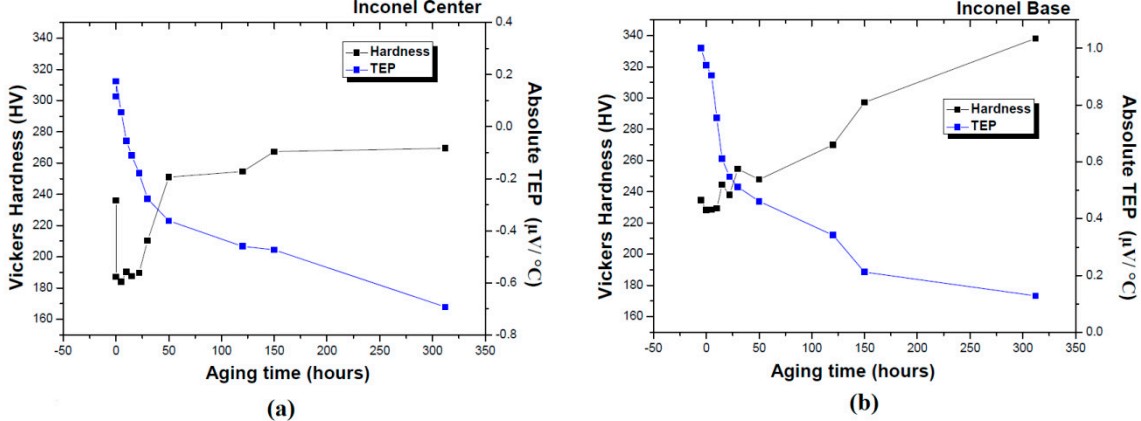

**Figure 12.** Graphs for comparison of Vickers hardness vs. aging time in the Inconel 825 alloy for (**a**) the central zone and (**b**) the base zone.

## 4. Conclusions

The precipitation behavior in a clad pipe (API X65/Inconel 825) has been studied using thermoelectric power and hardness measurements. The study clearly revealed that the most important microstructural parameters probably affecting the electron flux were the precipitation process and the grain size/shape. It has been shown that the major cause to the decrease of TEP in the aging process was the depletion or enrichment of the matrix from the precipitating elements and anisotropy due to different microstructures at the distinct (API X65/Inconel 825) zones. For the case of micro-alloyed X65 steel, the decrease in TEP for the different aging times is attributed to the precipitation of the nanoparticles ($Fe_3C$) and carbide-$\varepsilon$ ($Fe_2C$). For the case of Inconel 825 alloy, the decrease in TEP for the different aging times is attributed to the precipitation of $MC_6$ carbides and by the formation of $M_{23}C_6$ and MC precipitates. This study has also shown that the rate of decrease of the TEP value of aged samples at the interface region was greater than that of the SHT sample. This was expected and is consistent with previous studies [4,18], which show that the TEP variations are mainly related to the increase in solid solution content of carbon. It is concluded that the thermoelectric potential technique (TEP) is very sensitive to microstructural changes. It can be used reliably for the evaluation or monitoring of precipitation in aging processes of bonded clad pipes. This work also suggests that the TEP measurements, which are easy and rapid to make, could be used effectively to determine the condition of the clad-base metal interface.

**Author Contributions:** Methodology, H.C. and R.C.; investigation, H.C., M.L.C., R.C., and M.S.; writing—review and editing; H.C., M.L.C., R.C., and P.H.; supervision, H.C.; project administration, H.C.

**Acknowledgments:** This work was performed at UMSNH-MEXICO with partially funding from CONACYT-MEXICO under project CB-2015/256013 and Prodep/CA-140.

**Conflicts of Interest:** The authors declare no conflict of interest.

## Nomenclature

| | |
|---|---|
| TEP | Thermoelectric Potential |
| SEM | Scanning Electron Microscopy |
| CRA | Corrosion Resistant Alloy |
| HSLA | High-Strength Low Alloy Steel |
| HV | Vickers Hardness |
| EDS | Energy Dispersive Spectroscopy |
| SHT | Solution Heat Treatment |

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
