# Peer review of "Evaluation of the Precipitation Process of a Clad Pipe by the Thermoelectric Potential Technique"

_metals, doi:10.3390/met9121274_

Round 1

Reviewer 1 Report

Thank you for addressing the comments.

Author Response

SIGNIFICANT REVISIONS AND NEW DATA WERE ADDED IN THE MANUSCRIPT TO FURTHER CONSIDERATION FOR PUBLICATION OF THE MANUSCRIPT IN METALS. ALL CHANGES WERE HIGHLIGHTED

THANK YOU FOR YOU TIME AND CONSIDERATION

BEST REGARDS

HECTOR CARREON ET. AL.

Reviewer 2 Report

I have a few comments:

The description of the vertical axis in the figure should be corrected, the Vicker HV scale, -  HV is not a unit !!!!
In addition, there is no information under what load (force) the relied hardness measurement was - e.g. HV 0.01 HV or HV 0.1?
Axis description should be: Hardness Vickers HV .... (load/force), -
The authors write in the text about hardness and the second time about microhardness - please correct and harmonize the descriptions

Author Response

The description of the vertical axis in the figure should be corrected, the Vicker HV scale, -  HV is not a unit !!!!

RESPONSE: THE FIGURES 3,9,11 and 12 WERE MODIFIED WITH THE REVIEWER COMMENTS  

In addition, there is no information under what load (force) the relied hardness measurement was - e.g. HV 0.01 HV or HV 0.1?
Axis description should be: Hardness Vickers HV .... (load/force), -

RESPONSE: The indenter (made of diamond) has a square-base pyramidal geometry with an included angle of 136°. The Vickers hardness number (HV) is the ratio of applied load to the surface area of the indent. According to the Vickers prescription HV = 1.8544P/d2, and the result is reported as the average value in units of kg/mm2.

The authors write in the text about hardness and the second time about microhardness - please correct and harmonize the descriptions

RESPONSE: WE CORRECTED THE HARDNESS DESCRIPTION  THROUGHOUT THE MANUSCRIPT

FINALLY, SIGNIFICANT REVISIONS AND NEW DATA WERE ADDED IN THE MANUSCRIPT TO FURTHER CONSIDERATION FOR PUBLICATION OF THE MANUSCRIPT IN METALS. ALL CHANGES WERE HIGHLIGHTED

THANK YOU FOR YOU TIME AND CONSIDERATION

BEST REGARDS

HECTOR CARREON ET. AL.

This manuscript is a resubmission of an earlier submission. The following is a list of the peer review reports and author responses from that submission.

Round 1

Reviewer 1 Report

This is an interesting bit of work that can change the way we perceive NDT. However, the written English needs further improvement. The explanation of certain physical changes are not clear. The thermoelectric measurements are not explained clearly. The control variables are not stated and the data obtained has not been elucidated. Please find some corrections in the attached file.

Reviewer 2 Report

EDS data should be provide to study the composition. Thermoelectric potential could be impacted by many other factors, such as doping level. It is not reliable to conclude that the change in thermoelectric potential is caused by the precipitates.

Reviewer 3 Report

The paper deals the microstructural change of a steel/Ni alloy clad during aging treatment. The paper is not possible to include the journal due to the following reasons.

The property of clad depends on the quality of bonding at the interface but the paper is not discussed well in this issue. Even the method to prepare the clad is not demonstrated well. The authors mentioned “rold cladding technique”, but the process is not well known. Condition not mentioned. The authors measured diffusion layer as found in Fig. 5 through SEM observation only. No phase analysis was made. The following phase analysis (Figs.6 & 7) is not well made just based on the references.